# Simulated Microgravity Inhibits the Proliferation of Chang Liver Cells by Attenuation of the Major Cell Cycle Regulators and Cytoskeletal Proteins

**DOI:** 10.3390/ijms22094550

**Published:** 2021-04-27

**Authors:** Chi Nguyen Quynh Ho, Minh Thi Tran, Chung Chinh Doan, Son Nghia Hoang, Diem Hong Tran, Long Thanh Le

**Affiliations:** 1Animal Biotechnology Department, Institute of Tropical Biology, Vietnam Academy of Science and Technology, Ho Chi Minh 700000, Vietnam; quynhchihonguyen@gmail.com (C.N.Q.H.); doanchinhchung@gmail.com (C.C.D.); hoangnghiason@yahoo.com (S.N.H.); 2Biotechnology Department, Graduate University of Science and Technology, Vietnam Academy of Science and Technology, Ha Noi 100000, Vietnam; minh.tt@vlu.edu.vn; 3Technology Department, Van Lang University, Ho Chi Minh 700000, Vietnam; 4Children Institute Research, UT Southwestern Medical Center, 6000 Harry Hines Blvd., suite NL11120C, Dallas, TX 75390, USA; DiemH.Tran@utsouthwestern.edu

**Keywords:** Chang Liver Cells, cell cycle regulators, cytoskeleton, proliferation, simulated microgravity

## Abstract

Simulated microgravity (SMG) induced the changes in cell proliferation and cytoskeleton organization, which plays an important factor in various cellular processes. The inhibition in cell cycle progression has been considered to be one of the main causes of proliferation inhibition in cells under SMG, but their mechanisms are still not fully understood. This study aimed to evaluate the effects of SMG on the proliferative ability and cytoskeleton changes of Chang Liver Cells (CCL-13). CCL-13 cells were induced SMG by 3D clinostat for 72 h, while the control group were treated in normal gravity at the same time. The results showed that SMG reduced CCL-13 cell proliferation by an increase in the number of CCL-13 cells in G0/G1 phase. This cell cycle phase arrest of CCL-13 cells was due to a downregulation of cell cycle-related proteins, such as cyclin A1 and A2, cyclin D1, and cyclin-dependent kinase 6 (Cdk6). SMG-exposed CCL-13 cells also exhibited a downregulation of α-tubulin 3 and β-actin which induced the cytoskeleton reorganization. These results suggested that the inhibited proliferation of SMG-exposed CCL-13 cells could be associate with the attenuation of major cell cycle regulators and main cytoskeletal proteins.

## 1. Introduction

Simulated microgravity (SMG) can be generated through several mechanisms, such as rotating wall vessels [1], 2D clinostats [2], 3D clinostats [3,4,5,6], or random positioning machines [7,8]. The model clinostat environment consists of purely rotational flow that is perpendicular to the gravitational field to interfere with the recognition of the gravitational acceleration vector of a biological system [9]. Clinostat with a rotating axis is called clinostat 2D. Meanwhile, in clinostat 3D, a second shaft is installed perpendicular to the first, operating at a constant speed and orientation. The slow rotation of clinostat could prevent the responses triggered by gravity [10]. These systems can modulate the orientation of the research object, in which this object cannot perceive the gravitational acceleration vector [11]. Previous studies have reported that microgravity inhibits the proliferation of several cell lines, such as human hematopoietic progenitor cells [12], bone marrow mesenchymal stem cells [13], and mouse mesenchymal stem cells [14]. SMG was also reported inducing markedly changes in cytoskeleton of many cell types. The remodeling of tubulin has been demonstrated in endothelial cells under microgravity conditions [15]. F-actin reorganization inhibited the migration of rat mesenchymal stem cells induced by SMG [16]. Breast cancer MDA-MB-231 cells exposed to SMG showed a dramatic reorganization in cytoskeleton, resulting in a different configuration compared to the control group [17]. Neural crest stem cells under SMG condition showed a disrupted organization of filamentous actin and increased globular actin level [18]. The MCF-7 cells showed a rearrangement of the F-actin and tubulin with holes and accumulations in the tubulin network [19].

A recent study showed that microgravity induced lipid dysregulation in the mouse liver [20], and SMG similarly affected rat liver function by altering the metabolism of loureirin B and the expression of major cytochrome P450 [21]. The liver is one of the largest organs in the human body and plays an important role in the metabolism of carbohydrates, proteins, and lipids [22]. Hepatocytes occupy approximately 80% of the liver volume [22], and the commonly used hepatocyte model is CCL-13 cells, which are functionally similar to normal liver cells [23]. Khaoustov et al. reported that the SMG created by the Rotary Cell Culture System is conducive to aggregation of primary human liver cells and the formation of liver tissue-like structures [24]. In addition, the incorporation of hepatocyte spheroids was enhanced within the scaffolds by providing microgravity conditions [25]. The low fluid shear microgravity environment may increase tissue-like self-organization of primary porcine hepatocyte [26]. Pharmacokinetic rate of drug metabolism by hepatocytes is promoted in model SMG [27]. Human hepatic cell line (HepG2) showed the lower transcription of CYP3A4 marker and higher consumption of amino acids and release of ketoacids and formiate under SMG condition [28]. These modifications could be caused by the changes in cell proliferation and structure, which are sensitive to SMG. However, the effects of SMG on the cytoskeleton and proliferative ability of human liver cells have not been well characterized. The objective of the present study was to evaluate the effects of SMG on the proliferation and cytoskeleton of hepatocyte. CCL-13 cells were used as a model to assess the proliferation, demonstrating by cell viability, cell cycle progression, and expression of major cell cycle-related regulators. Moreover, we evaluated morphological changes in the nucleus and cytoplasm of CCL-13 cells, with a focus on the expression of prominent structural proteins and the reorganization of microtubule and microfilament bundles.

## 2. Results

### 2.1. Proliferation of Chang Liver Cells under SMG

The number of CCL-13 cells was counted using a Cytell Microscope (Appendix A). The cell number in the control group in the 3-day culture increased to 10.1 × 10^3^ cells/well, which was 5-fold higher than the original seeding number (2 × 10^3^ cells/well) (Figure 1A). The number of CCL-13 cells in the SMG group (7.09 × 10^3^ cells/well) was lower than that of the control group over the 3-day culture.

The WST-1 assay was also employed to assess CCL-13 cell proliferation (Appendix A). The absorbance value of CCL-13 cells in the control group in the 3-day culture was 0.76 ± 0.01, which was higher than that of cells in the SMG group (0.62 ± 0.03) (Figure 1B). These results indicated that CCL-13 cells from the SMG group exhibited lower proliferation than cells from the control group.

The cycle progression of CCL-13 cells was evaluated by flow cytometry (Appendix A). The ratio of SMG cells in the G0/G1 phase was higher than the control cells (90.60 ± 0.40% vs. 86.93 ± 0.23%, respectively) (Figure 1C). The percentage of CCL-13 cells in the S phase and G2/M phase was higher in the control cells than the SMG cells. These data revealed that SMG conditions resulted in CCL-13 cells moving to the cell cycle arrest phase.

### 2.2. Effect of SMG on Cell Cycle Regulators

The Western blot results (Appendix A) indicated that the CCL-13 cells from the control group had a higher expression of cyclin A1 and A2 protein than cells in the SMG group (Figure 2). Downregulation of cyclin D1 was also observed in CCL-13 cells in the SMG group. Furthermore, CCL-13 cells in the SMG group showed reduced expression of Cdk 6, compared to cells in the control group. However, there was no difference in Cdk4 expression between the control and SMG cells (Figure 2).

### 2.3. Apoptosis Analysis

Flow cytometry analysis (Appendix A) demonstrated that CCL-13 cells in both groups exhibited similar percentages of viability and apoptosis (Figure 3A). Moreover, the nuclear morphology of CCL-13 cells in both groups showed normal morphology with no fragmentation (Figure 3B).

### 2.4. Morphological Evaluation of CCL-13 Cells under SMG Conditions

The effects of SMG on CCL-13 cell proliferation were further estimated by morphological evaluation. The FSC value of CCL-13 cells in the SMG group was higher than that of cells in the control group (9.80 × 10^6^ vs. 8.78 × 10^6^, respectively) (*p* < 0.001) (Figure 4A and Appendix A), suggesting that the diameter of CCL-13 cells in the SMG group was greater than that of cells in the control group. The nuclear area of CCL-13 cells in the SMG group was lower than that of cells in the control group (244.50 ± 2.79 µm^2^ vs. 254.75 ± 1.41 µm^2^, respectively) (*p* < 0.01) (Figure 4B and Appendix A). The parameter of nuclear generated by the Cytell microscope is nuclear form factor (1.0 = circle, <1.0 = non-circular), which evaluates nuclear integrity; there was no difference in the nuclear form factor in CCL-13 cells in both groups (Figure 4C and Appendix A).

### 2.5. Effects of SMG on Cytoskeletal Protein Expression

In order to clarify the effects of SMG on the expression of cytoskeletal proteins, Western blotting and immunofluorescence staining were performed to assess the modification of microfilaments and microtubules in CCL-13 cells. As seen in the Figure 2, CCL-13 cells in the SMG group exhibited downregulated beta-actin and alpha-tubulin 3, compared to cells in the control group. Figure 5 shows the distribution of microtubules in CCL-13 cells. CCL-13 cells in the control group showed perinuclear accumulation of microtubules, while microtubules in SMG cells were evenly distributed around the nucleus.

Actin filament staining demonstrated that the cytoplasm in cells in the control group presented thicker microfilament bundles than cells in the SMG group (Figure 6). The microfilament formation in CCL-13 cells in the control group was higher than that in the SMG-induced CCL-13 cells. The microfilament bundles were distributed in parallel and spread through the length of CCL-13 cells in the control group, while CCL-13 cells in the SMG group displayed crossed thin microfilament bundles in the cytoplasm.

## 3. Discussion

In the current study, the density of CCL-13 cells in the control and SMG groups was 5-fold and 3.5-fold higher than seeding density, respectively. In addition, the WST-1 assay showed a lower proliferation rate in SMG-induced CCL-13 cells compared to control cells. These results suggested that SMG attenuated in vitro CCL-13 proliferation. The reduced proliferation rate in cells does not result from enhanced cell death under SMG conditions [4], and could be due to other mechanisms, such as cell cycle progression [29]. For example, the inhibited proliferation of SMG-induced bone marrow mesenchymal stem cells was related to the blocking of the cell cycle in G2/M [13]. Another study showed that tumor cell growth was dramatically inhibited under SMG conditions by reverting to catabolic metabolic machinery for housekeeping functions [30]. Plett et al. reported that the slowed proliferation of CD34+ bone marrow cells was caused by a prolonged S phase [12]. In the present study, we found that the inhibited proliferation of SMG-induced CCL-13 cells was not only governed by changes in cell cycle progression but also modulated by changes in the expression of cell cycle-related regulators, including cyclin A1, cyclin A2, Cdk4, and Cdk6. Flow cytometry analysis indicated that the percentage of CCL-13 cells was decreased in the G2/M phase and increased in G0/G1 phase, revealing that the reduced proliferation of SMG-induced CCL-13 cells was associated with the transition to arrest phase.

Cell proliferation requires an increase in cell division, which is controlled by cell cycle regulators such as cyclins and Cdks [31]. Cyclin A1 and A2 play important roles in the S phase and G2/M transition in all proliferating cells [32,33,34,35]. In the G1 phase, enhanced cyclin D synthesis promotes the association of Cdk 4/6 and cyclin D complexes that control the progression through G1 and G1/S transition [36,37,38]. Thus, changes in the expression of Cdk 4, Cdk 6, and cyclin D lead to alterations in the cell cycle. In the current study, the downregulated expression of cyclin D correlated with the reduced expression of Cdk6 in SMG-induced CCL-13 cells, leading to the G1 phase delay as observed in SMG-induced CCL-13 cells. The downregulated expression of cyclin A1 and cyclin A2 was also observed in SMG-induced CCL-13 cells, which resulted in the low percentage of SMG-induced CCL-13 cells in the S phase. Moreover, there was no difference in viability and apoptosis in CCL-13 cells between the control and SMG groups, which is consistent with that of findings of the Benavides DammT group [4]. These results suggested that the mechanism of inhibited proliferation was caused by the promotion of the cell cycle arrest phase, which was caused by the downregulation of major cell cycle regulators.

In cell cycle progression, duplication of genomic DNA and other cellular components generates nucleus reconstruction. At the interphase, nuclear volume increases approximately twice in dividing cells before reaching the final size [39,40]. In the current study, the average nuclear size of CCL-13 cells in the control group was higher than that of cells in the SMG-group. Therefore, the number of dividing cells under SMG conditions was less than that of cells under control conditions, which resulted in the inhibited proliferation of CCL-13 cells.

The cytoskeleton is a special structure that contributes to three functions, namely organizing cell structures, supporting cell movement and changing shape, and modulating cell–cell and cell–environment interactions [41]. Cytoskeletons include microtubules, intermediate filaments, and microfilaments, in which microtubules and microfilaments are essential for cell division [42]. Microtubules are formed by the polymerization of tubulin. During cell division, microtubules form spindles that separate sister chromatids into two cells [43]. Actin is a component of microfilaments that contributes to the formation of the cleavage furrow during cell division [44]. The changes in actin and tubulin synthesis could modify the polymerization and formation of microfilaments and microtubules. In this study, the synthesis of β-actin and α-tubulin 3 was reduced in SMG-induced CCL-13 cells, indicating a decrease in the formation of supporting structures for cell division. Thus, downregulation of β-actin and α-tubulin could lead to the inhibition of cell proliferation.

Cyclin D1 and Cdk 4/6 contribute to the organization of actin filaments and microtubules [45,46]. A recent study demonstrated that cytoskeleton organization was modulated by the interaction of Cdk6 with a number of cytoskeleton-related proteins [36]. Cdk6 contributes to the regulation of a panel of genes associated with (de-)polymerization of actin [47]. Cdk6-deficient cells have been shown to diminish the formation of F-actin, which is polymerized to construct microfilaments [47]. In the present study, SMG-induced CCL-13 cells showed downregulated expression of Cdk6, which correlated with the reduced expression of β-actin and impaired microfilament formation in the cytoplasm. This consequently reduced cell motility and division.

In conclusion, our study demonstrate that SMG induces the reduction in the main cell cycle regulators and attenuation of cytoskeletal proteins CCL-13 cells, resulting in the inhibited proliferation of CCL-13 cells. The further research should be conducted to evaluate the changes of hepatocyte function, including metabolism, detoxification, and protein synthesis to clarify the effects of SMG on human liver function.

## 4. Materials and Methods

### 4.1. Cell Culture

CCL-13 cells were cultured in DMEM/Ham’s F-12 (DMEM-12-A, Capricorn Scientific, Ebsdorfergrund, Germany) supplemented with 15% FBS (FBS-HI-22B, Capricorn Scientific, Ebsdorfergrund, Germany) and 1% Pen/Strep (15140-122, Gibco, Thermo Fisher Scientific, Inc., Waltham, MA, USA). To simulate microgravity, CCL-13 cells were seeded in T-25 flasks and 96-well plates at a defined density of 1 × 10^5^ cells/flask and 2 × 10^3^ cells/well, respectively. Then, the T-25 flasks and 96-well plates were carefully filled with culture medium without bubbles to avoid the shearing of fluid [48]. The flasks and plates were mounted to the stage of inner frame of 3D clinostat (MiGra-ITB, Vietnam Academy of Science and Technology, Ha Noi, Vietnam), which was placed in a CO_2_ incubator (Sanyo MCO-18AIC CO_2_ Incubator, Sanyo Electric Co., Osaka, Japan) (Appendix A). CCL-13 cells were exposed to SMG using a 3D clinostat for 72 h. The control group was treated at 1 g in the same CO_2_ incubator.

### 4.2. Cell Density Measurement

CCL-13 cells were cultured in 96-well plates (161093, Thermo Fisher Scientific, Inc., Waltham, MA, USA) at a density of 2 × 10^3^ cells/well. Each well was filled with 395 µL of culture medium, and parafilm tape was used to cover the wells. CCL-13 cells were subjected to SMG for 72 h, and the cell culture was thereafter removed. The nuclei were stained with Hoechst 33342 (14533, Sigma-Aldrich, Munich, Germany) for 15 min. The cells were washed three times with phosphate-buffered saline (PBS; Gibco, Thermo Fisher Scientific, Inc., Waltham, MA, USA). The number of CCL-13 cells was determined using the Cell Cycle App. of a Cytell Microscope (GE Healthcare, Arlington Heights, IL, USA).

### 4.3. WST-1 Cell Viability Assay

CCL-13 cells were cultured in 96-well plates (161093, Thermo Fisher Scientific, Inc., Waltham, MA, USA) at a density of 2 × 10^3^ cells/well. Each well was filled with 395 µL of culture medium, and parafilm tape was used to cover the wells. CCL-13 cells were subjected to SMG for 72 h; after 72 h, the culture medium was removed and 100 µL fresh medium and 10 µL WST-1 (11644807001, Roche, Basel, Switzerland) were added to each well. Cells were incubated for 3.5 h at 37 °C in a 5% CO_2_ atmosphere. The optical density (O.D.) was measured at 450 nm using a GloMax^®^ Explorer Multimode Microplate Reader (Promega Corporation, Fitchburg, WI, USA). Control cells were not subjected to SMG.

### 4.4. Flow Cytometry Analysis

CCL-13 cells were cultured in T-25 flasks (160430, Thermo Fisher Scientific, Inc., Waltham, MA, USA) at a density of 1 × 10^5^ cells/flask. Flasks were filled with 80 mL of culture medium, and the filter-cap was slowly screwed to avoid the formation of bubbles. CCL-13 cells were subjected to SMG for 72 h. Flow cytometry analysis was performed using the FITC Annexin V Apoptosis Detection Kit I (556547, BD Biosciences, San Jose, CA, USA) in BD Accuri C6 Plus (BD Biosciences, San Jose, CA, USA). To analyze cell cycle progression, CCL-13 cells were fixed with 4% paraformaldehyde (09154-85, Nacalai Tesque, Kyoto, Japan) for 15 min. CCL-13 cells were washed twice with cold PBS and resuspended in 1X Binding Buffer at a concentration of 1 × 10^6^ cells/mL. CCL-13 cells were stained with 5 µL PI (51-66211E, BD Biosciences, San Jose, CA, USA). Cell cycle analysis was performed by measuring the cellular DNA content using a flow cytometer.

### 4.5. Western Blot Analysis

CCL-13 cells were harvested from T-25 flasks, and the lysate was prepared with Optiblot LDS Sample Buffer (ab119196, Abcam, Cambridge, MA, USA). Equal amounts of protein (10 µg/well) were loaded into the wells of the Precast Gel SDS-PAGE 4–12% (ab139596, Abcam, Cambridge, MA, USA). The gel was run in Optiblot SDS Run Buffer (ab119197, Abcam, Cambridge, MA, USA) for 2 h at 50 V. The protein was transferred to a PVDF membrane (ab133411, Abcam, Cambridge, MA, USA), and the membrane was blocked overnight at 4 °C with blocking buffer (ab126587, Abcam, Cambridge, MA, USA). The membrane was incubated with primary antibodies in blocking buffer overnight at 4 °C. Anti-beta-actin (ab8226, Abcam, Cambridge, MA, USA) and anti-alpha-tubulin (ab52866, Abcam, Cambridge, MA, USA) were used at a 1:10,000 dilution. Anti-cyclin A1 + cyclin A2 (ab185619, Abcam, Cambridge, MA, USA), anti-Cdk4 (ab137675, Abcam, Cambridge, MA, USA), and anti-Cdk6 (ab124821, Abcam, Cambridge, MA, USA) were used at a 1:5000 dilution. Anti-GAPDH (ab181602, Abcam, Cambridge, MA, USA) was used as the control, at a 1:10,000 dilution. The membrane was washed three times with TBST for 10 min each. The membrane was incubated with secondary antibody in blocking buffer at room temperature for 1 h. Goat anti-mouse IgG (HRP) (ab6789, Abcam, Cambridge, MA, USA) and goat anti-rabbit IgG (HRP) (ab6721, Abcam, Cambridge, MA, USA) were used to detect the beta-actin antibody and other primary antibodies, respectively. The blots were visualized using the ECL Western Blotting Substrate Kit (ab65623, Abcam, Cambridge, MA, USA). Imaging was carried out with an X-ray film.

### 4.6. Microtubule Staining

CCL-13 cells were cultured in 96-well plates at a density of 2 × 10^3^ cells/well, in 395 µL/well culture medium. Microtubules were stained with 50 nM SiR-tubulin/well (CY-SC002, Cytoskeleton, Inc., Denver, CO, USA). CCL-13 cells were subjected to SMG for 72 h, and the structure of microtubule bundles was evaluated under a Cytell microscope (GE Healthcare, Chicago, IL, USA).

### 4.7. Microfilament Staining and Nuclear Morphology Evaluation

CCL-13 cells were cultured in 96-well plates at a density of 2 × 10^3^ cells/well and subjected to SMG for 72 h. CCL-13 cells were fixed with 4% paraformaldehyde (Nacalai Tesque, Kyoto, Japan) for 30 min, before being permeabilized with 0.1% Triton X-100 (Merck, Darmstadt, Germany) overnight at 4 °C. Phalloidin CruzFluor™ 488 Conjugate (sc-363791, Santa Cruz Biotechnology, Santa Cruz, CA, USA) was used to stain actin filaments for 1 h. The nuclei were stained with Hoechst 33342 (14533, Sigma-Aldrich, Munich, Germany) for 15 min. The cells were washed three times with PBS (Gibco, Thermo Fisher Scientific, Inc., Waltham, MA, USA) for 10 min each. The structure of the microfilament bundles was evaluated under a Cytell microscope, and the Cell Cycle App was used to derive the nuclear area, nuclear intensity, and nuclear shape value [49].

### 4.8. Statistical Analysis

The data were analyzed for statistical significance by one-way ANOVA, where *p* < 0.05 was considered statistically significant.

## Figures and Tables

**Figure 1 ijms-22-04550-f001:**
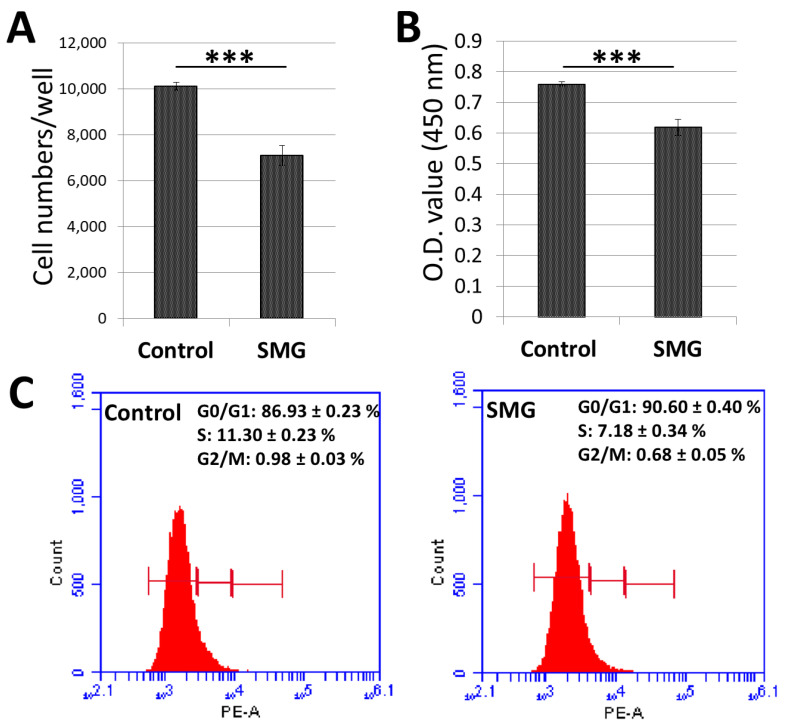
Proliferation of Chang Liver Cells (CCL-13) cells in control and simulated microgravity (SMG) groups. (**A**) The number of CCL-13 cells was counted using the Cell Cycle App. of the Cytell microscope (*n* = 12). (**B**) CCL-13 cell proliferation was assessed by WST-1 assay (*n* = 12). (**C**) Cycle progression of CCL-13 cells was analyzed by flow cytometry (*n* = 4). *** indicates significant difference compared with the control group (*p* < 0.001).

**Figure 2 ijms-22-04550-f002:**
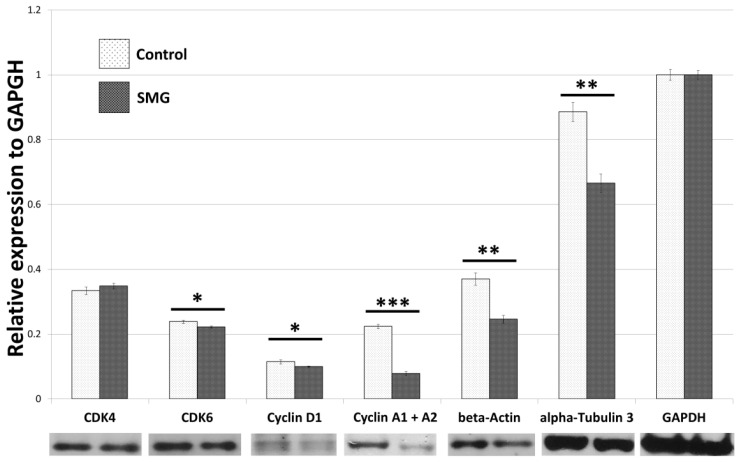
Western blot analysis of major cell cycle regulators and main cytoskeletal proteins in CCL-13 cells (*n* = 3). *** indicates significant difference compared with the control group (*p* < 0.001); ** indicates significant difference compared with the control group (*p* < 0.01); * indicates significant difference compared with the control group (*p* < 0.05).

**Figure 3 ijms-22-04550-f003:**
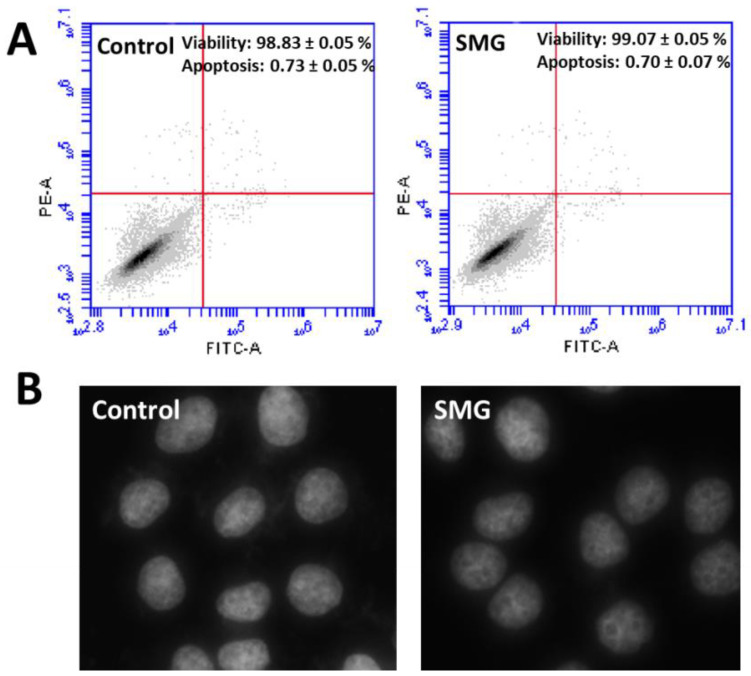
The apoptosis and viability of CCL-13 cells. (**A**) Flow cytometry analysis of apoptosis and viability in CCL-13 cells (*n* = 4). (**B**) Nuclear morphology of CCL-13 cells.

**Figure 4 ijms-22-04550-f004:**
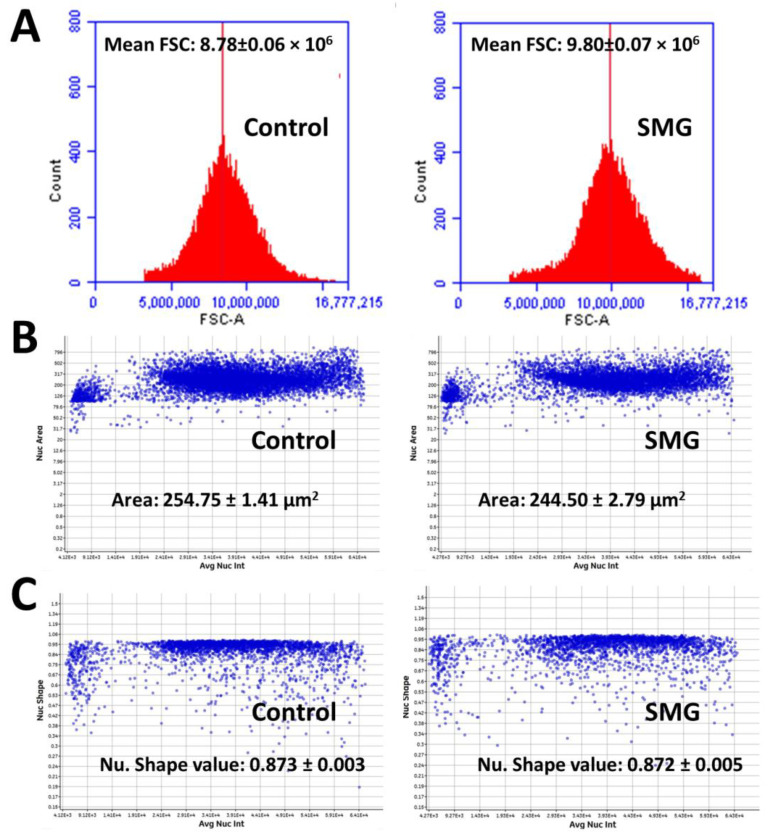
Morphology of CCL-13 cells. (**A**) The mean forward scatter (FSC) value indicates the diameter of CCL-13 cells (*n* = 5). (**B**) The distribution of CCL-13 nuclear area in relation to the total nuclear intensity (*n* = 12). (**C**) The distribution of nuclear shape value in relation to the total nuclear intensity (*n* = 12).

**Figure 5 ijms-22-04550-f005:**
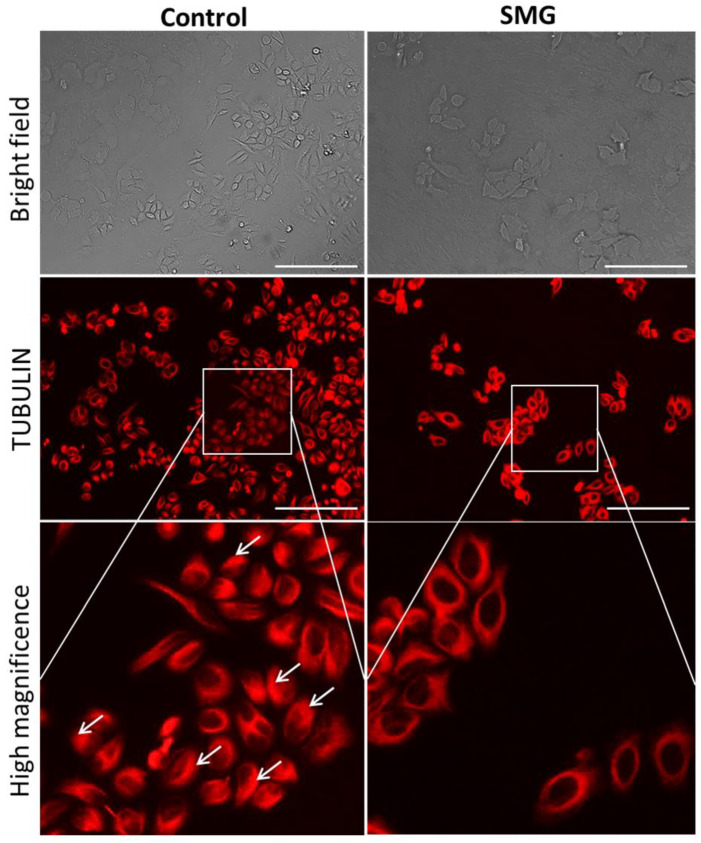
Distribution of microtubules in CCL-13 cells. Microtubules were stained with Sir-Tubulin (red). The white arrows indicate perinuclear accumulation of microtubules. Scale bar = 223.64 µm.

**Figure 6 ijms-22-04550-f006:**
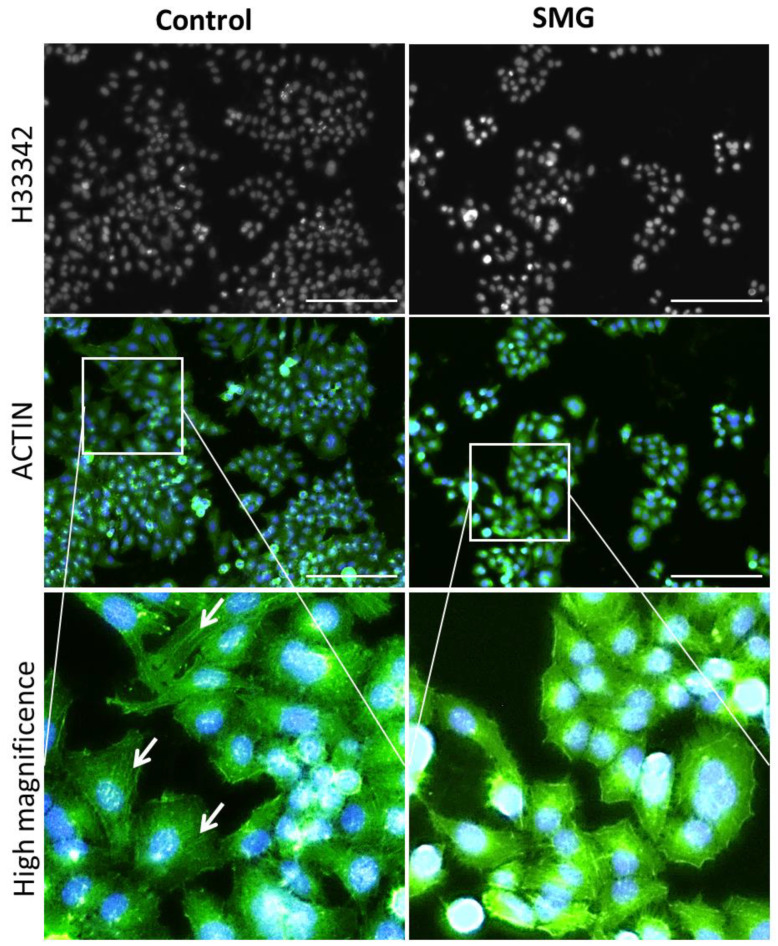
Distribution of microfilament bundles in CCL-13 cells. Nuclei were counterstained with H33342, and microfilaments were counterstained with Phalloidin (green). The white arrows indicate microfilament bundles. Scale bar = 223.64 µm.

## Data Availability

Not applicable.

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
