# Peer review of "Simulated Microgravity Inhibits the Proliferation of Chang Liver Cells by Attenuation of the Major Cell Cycle Regulators and Cytoskeletal Proteins"

_ijms, 2021, doi:10.3390/ijms22094550_

Round 1

Reviewer 1 Report

The article “ Simulated Microgravity Inhibits the Proliferation of Chang Liver Cells by Attenuation of the Major Cell Cycle Regulators and Cytoskeletal Proteins, opportunities  by  Chi Nguyen Quynh Ho et al. is interesting. 

The authors to present the results of simulated microgravity (SMG) on the proliferative ability and cytoskeleton changes of Chang Liver Cells (CCL-13).

However, several points have to be addressed:

Abstract, Introduction, Results and Discussion section need to be rewrite.

All data should be described in more details with the number of independent experiments.

Abstract too short and does not contain sufficient information. Please, provide more information (a reason, time, conclusion...).

Introduction

Introduction is not covered enough and should be expanded. Please, provide more information. The aim of the study is not entirely clear.

  1. Line 31-33. Move or delete the text "Previous studies have reported..."    In the in Discussion same text and the References.

Results

The number of independent experiments should be indicated in each Figure legend.

  1. Line 55-56. Results are shown only for 72 h. Are there any data at the different time points? One time point is not enough to compare the data.
  2. Line 63-64. Figure1. A, B – Please indicate the number of independent experiments.
  3.  Fig 1C. Data is it not so different. How many experiments (n=?), duplicate were analyzed? The quantification of the cell cycle populations not described. It is difficult to interpret.
  4. Line 72. Authors should include the results of cell cycle regulators in the separate section (2.2.)
  5. Line 78. Figure 2. Please indicate the number of independent experiments. 
  6. Line 83-85. 2.2. Morphological evaluation of CCL-13 cells under SMG conditions. This section need to be rewrite. Authors should include the results of morphology. Flow cytometry data may to move in the separate section (or in the section 2.1.). 
  7. Line 89. Figure 3. – No statistical data. It is difficult to interpret. Please state in more details. Indicate time point.
  8. Line 101. Figure 4. – No statistical data. Please indicate the number of independent experiments. It is difficult to interpret.
  9. Line 104. – 2.3. Effects of SMG on cytoskeleton-related gene expression … do you mean proteins  expression?

Discussion

Authors measure the effects of SMG on the cytoskeleton and cell proliferation, but neither gives a reason why they do this. Please state in more details.

   11. Line 143-144. The cell cycle profiles between groups are not so different. 

   12. Line 188-189. Please make conclusions in the discussion section.

Methods

13. Line 197. - Please describe 3D microgravity simulation in more details or indicate the reference. Authors should move the describe of clinostat in the separate section. 

Please full expand all abbreviations: FCS, WST-1, SDS-PAGE, TBST…

Reviewer 2 Report

The authors have summitted an interesting manuscript, looking at proliferative changes from the angle of normal tissue cells rather than cancer cells. This is a nice approach to get further insights into the biological changes that space travelers are facing and undergoing.  The manuscript is certainly of interest to the microgravity research community and beyond.

The experiments are scientifically sound, and the results demonstrate the described results. The manuscript would gain on impact by extending and broadening it in several areas, some points are detailed below.

  1. Introduction
  • Please add a brief overview/description about the various methods by which SMG can be generated rather than just mentioning them.
  • Please extend your section of the current knowledge of cytoskeletal reorganization under SMG, and possible implications to the three-dimensional growth of tissues.
  • Please add the current knowledge of the impact of SMG on other cells, from both regular human tissues and liver tumor cells.
  1. Results
  • Line 66: as you describe an attenuation of cell cycle regulators due to SMG, please explain the nature of the cell cycle arrest phase (temporary?).
  • Figure 2: please indicate the amount of protein loaded per lane (section 4 describes “equal amounts”).
  1. Discussion
  • The presented results are all reasonably discussed. What is missing is the discussion about the broader implications of the cell cycle attenuation on the liver tissue as well as an outlook how these findings can support future studies.
  1. Materials and Methods
  • Please add a separate sub-section for the experimental setup and procedure on the 3D-clinostat, since it is a critical method for the entire study.
  • Please review this section to make sure all instruments and supplies have the manufacturers/distributors listed (example: 4.3. – WST-1 and 4.4 – flow cytometer).

Round 2

Reviewer 1 Report

The manuscript has been significantly improved and now warrants publication in IJMS.

Two comments:

Line  16-17. - “cell proliferation” repeated twice in the same sentence. Please to rephrase the sentence. (e.g. "which plays an important factor in various cellular processes".)

Line 323: Please full expand abbreviation: SMG...

Nevertheless, I hope the authors will extend the experiment time in the further research to clarify the effects of SMG on cell cycle progression.

Author Response

Dear Prof. Dr. Maurizio Battino and Editorial Board,

Thank you so much for your consideration of our manuscript. We would like to thank the Reviewer(s) for careful and thorough reading of the manuscript and for the thoughtful comments and constructive suggestions, which help to improve the quality of this manuscript. Each comment has been carefully considered point by point and responded. Responses to the reviewers and changes in the revised manuscript are as follows (the reviewer’s comments are in italics). The changes of our manuscript are highlighted in yellow.

Response to Reviewer 1 Comments

Comment 1. Line 16-17. - “cell proliferation” repeated twice in the same sentence. Please to rephrase the sentence. (e.g. "which plays an important factor in various cellular processes".)

Response 1. Thank you so much for your comment. The phrase “contributing to cell proliferation” was corrected to “various cellular processes”. (Page 1, line 16)

Comment 2. Line 323: Please full expand abbreviation: SMG...

Response 2. We have added more abbreviation in abbreviation section. (Page 11,  the abbreviation are highlighted in yellow)

Comment 3. Nevertheless, I hope the authors will extend the experiment time in the further research to clarify the effects of SMG on cell cycle progression.

Response 3. In the further research, we will extend the experiment time to clarify the effects of SMG on cell cycle progression, structure and function in CCL-13 cells and other cell lines.

We hope that our corrections could meet your requirements,

Thank you so much.
